# Confirmation of CD19+ B-Lymphocyte Depletion Prior to Intake of the Second Dose of Ocrelizumab in Multiple Sclerosis Patients

**DOI:** 10.3390/biomedicines11020353

**Published:** 2023-01-26

**Authors:** Marija Radmilo, Sanda Pavelin, Igor Vujović, Joško Šoda, Maja Rogić Vidaković

**Affiliations:** 1School of Medicine, University of Split, 21000 Split, Croatia; 2Department of Neurology, University Hospital of Split, 21000 Split, Croatia; 3Signal Processing, Analysis, and Advanced Diagnostics Research and Education Laboratory (SPAADREL), Faculty of Maritime Studies, University of Split, 21000 Split, Croatia; 4Laboratory for Human and Experimental Neurophysiology (LAHEN), Department of Neuroscience, School of Medicine, University of Split, 21000 Split, Croatia

**Keywords:** multiple sclerosis, ocrelizumab, immunophenotyping, B-lymphocytes, T-lymphocytes, NK cells, albumins, globulins

## Abstract

The aim of the retrospective study was to compare the immunophenotyping of T-lymphocytes, B-lymphocytes, and natural killer cells before the administration of the first and the second dose of ocrelizumab in 22 patients with multiple sclerosis in a three-year period (2019–2021) at the Department of Neurology of the University Hospital of Split. The values of cell immunophenotyping and protein electrophoresis, as well as laboratory parameters, were investigated. There was no significant decrease in serum albumin and globulins before the second dose of ocrelizumab (*p* > 0,05). A decrease in the number of T-lymphocytes before administration of the second dose of ocrelizumab was observed, but without statistical significance (*p* = 0.274). Significant depletion occurred in median CD19+ B-lymphocytes (*p* < 0.001) before the intake of the second dose of ocrelizumab confirming the primary action of ocrelizumab on the B cell lineage.

## 1. Introduction

### 1.1. Pathogenesis of Multiple Sclerosis

Multiple sclerosis (MS) is a chronic demyelinating disease of the central nervous system (CNS) with degenerative, inflammatory, and autoimmune etiology [1]. It is one of the most common non-traumatic, disabling diseases of young adults [2,3,4,5,6,7,8,9]. A classic pathological sign in the pathogenesis of MS is inflammatory lesions that become demyelinating plaques [10]. As a result of the inflammation, in addition to the damage to myelin, oligodendrocytes are also damaged. In the initial phase, axons are preserved, but as the disease progresses, irreversible axon damage occurs [11]. In the relapsing-remitting form of MS (RRMS), active lesions with central lymphocytic infiltration are present, while in the primary-progressive form of MS (PPMS), inactive lesions with active edges containing microglia and macrophages are more often present [12]. Although the therapeutic strategy targeting B-lymphocytes has recently attracted researchers, multiple sclerosis was traditionally considered a disease mediated by T cells, particularly CD4-positive T cells reactive to myelin antigens [13]. Additionally, increasing evidence suggests that macrophages derived from circulating monocytes and resident microglia play a pivotal role in the pathogenesis of multiple sclerosis [14].

In the pathogenesis of MS, altered CD4+ T-lymphocytes play an initial key role, although other cells of innate and acquired immunity are also pathologically altered [15,16,17,18]. CD4+ T- lymphocytes contain adhesive molecules on their surface that enable them to bind to the blood–brain barrier (BBB), and after binding, these activated lymphocytes secrete the matrix metalloproteinase (MMP) enzyme, which enables the cells to pass through the BBB into the CNS, where they encounter the antigen MBP (myelin basic protein) on microglia. In this way, pathological T-lymphocytes that secrete pro-inflammatory cytokines are reactivated, which further increases the permeability of BBB and promotes the formation of inflammatory lesions [17]. CD8+ T-lymphocytes are also the pathohistological finding of demyelinating changes [16]. CD8+ T-lymphocytes are the most responsible cells for the creation of demyelinating lesions in the brain through the Fas ligand (FasL) mechanism [19]. B-lymphocytes likewise damage myelin by producing antibodies that destroy it, but also by creating and maintaining follicle-like structures in inter-meningeal spaces, which primarily affect the appearance of subpial lesions and disease progression [20,21]. Plasma cells produce immunoglobulin G, i.e., oligoclonal bands (OCB), which are found in the cerebrospinal fluid of more than 95% of people with multiple sclerosis (pwMS). A positive finding of OCBs in the cerebrospinal fluid increases the risk of developing clinically isolated syndrome (CIS), and in addition to cerebrospinal fluid, they can be found in lesions [22,23]. Natural killer cells (NK) are also thought to play an important role in the immunopathogenesis of MS, through of still unknown mechanism [24,25,26,27,28,29,30,31].

### 1.2. Mechanism of Action of Disease-Modifying Ocrelizumab Drug

Ocrelizumab is a recombinant humanized intravenous anti-CD20 monoclonal antibody approved as first-line treatment for all PPMS patients and for RRMS patients with at least one relapse in the last two years or with at least one active lesion on MR [32]. It acts immune-suppressively by binding to the surface antigen CD20 on CD20+ B lymphocytes, called: pre-B-lymphocytes, mature, and memory cells [33,34]. It causes their depletion, while at the same time, it does not affect the decline of CD20- B lymphocytes, which include pro-B lymphocytes, stem cells, and plasma cells, thus saving the capacity for B lymphocyte reconstitution [32,35,36,37]. Selective depletion of CD20+ B lymphocytes is promoted by: apoptosis, antibody- and complement-dependent cellular cytotoxicity, and antibody-dependent cellular phagocytosis. In addition to CD20+ B lymphocytes, ocrelizumab also affects the depletion of CD19+ B lymphocytes, especially within two weeks of receiving the drug [38]. The median recovery of B-lymphocytes after depletion is 18 months [35], and the repopulation of B cells begins in the spleen and bone marrow, and then in the blood [39]. Ocrelizumab is considered a drug that acts exclusively on B-lymphocytes, but recent findings also point to its significant effect on T-lymphocytes [40]. The CD20 antigen is as well found in smaller amounts on T lymphocytes in the subset of CD3+CD20+ T cells, and it is known that these CD20+ T cells are a highly activated cell population [41]. The frequency of CD20+ T-lymphocytes is increased in patients with MS, and these cells have a pronounced pro-inflammatory phenotype with pathogenic properties [42]. So far, ocrelizumab has no known effect on NK cells [43,44]. Compared to interferon beta-1a in the OPERA I and OPERA II studies, ocrelizumab showed significantly better results. Furthermore, ocrelizumab is associated with a higher rate of improvement in disability than interferon beta-1a [45]. The most common side effect after ocrelizumab administration is an infusion reaction, followed by infections and laboratory abnormalities [46], delayed-onset neutropenia [47,48,49], a decrease in the total number of lymphocytes with a predominance of a decrease in CD8+ T-lymphocytes [50,51], and hypogammaglobulinemia [52,53,54]. Laboratory abnormalities of liver enzymes are not expected in patients with MS who regularly take ocrelizumab [55].

The present study’s aim was to compare the immunophenotyping of T-lymphocytes, B-lymphocytes, and NK cells before the administration of the first and second doses of ocrelizumab in pwMS treated at the Department of Neurology at the Clinical Hospital Center Split, Croatia in the three-year period (2019–2021). Investigation of T-lymphocytes, B-lymphocytes, and NK cells before the administration of the first and second doses of ocrelizumab serves to better understand the immunomodulatory effect of ocrelizumab.

## 2. Materials and Methods

### 2.1. Participants

A retrospective study initially enrolled a total of 24 adult pwMS treated with ocrelizumab in the period from 1 January 2019 to 31 December 2021, at the Department of Neurology at the Clinical Hospital Center, Split, Croatia (Figure 1). Two patients did not meet the inclusion criteria; overall, 22 pwMS were included in the study (Figure 1).

The exclusion criterion included the following: infusion reactions, skin infections, respiratory infections, known hypersensitivity to the active substance or one of the excipients, current active infection, severely immunocompromised patients, known active malignant diseases, and pregnancy. Infusion reactions can include symptoms ranging from itching and a rash, a headache or dizziness, trouble breathing, or abnormal heartbeat patterns. These reactions occurred mainly on the first infusion and were managed based on the particulars of the individual reaction.

After satisfying the clinical indication of active RRMS or PPMS, each patient underwent an extensive physical examination: ultrasound of the abdomen and breast, X-ray of the lungs, tumor markers (CEA—carcinoembryonic antigen, CA—carcinoma antigen, PSA- prostate-specific antigen, CYFRA—cytokeratin 21 fragments), quantiferon test, markers for hepatitis and HIV infection, positive varicella antibodies, and examination by a dermatologist (dermatoscopy) to rule out the presence of skin changes suspicious for malignancy. One of the patients in this period did not meet the inclusion criteria because a bladder tumor was diagnosed during the abdominal examination with a diagnosis of carcinoma in situ and a permanent risk of developing a malignant disease with the use of ocrelizumab. In another patient with PPMS, a melanocytic facial mole was removed before application, and the possibility of malignant disease was ruled out. None of the pwMS needed protective anti-tuberculosis therapy based on the quantiferon test, nor vaccination against varicella.

### 2.2. Collection of Data and Study Procedure

The collected data consisted of the demographic and clinical parameters including sex, age (years of the patient until the first administration of ocrelizumab), age groups (18–30 years, 30–40 years, 40–50 years, 50–60 years, and ≥60 years), the Expanded Disability Status Scale (EDSS) score, the average duration of the disease from the year of diagnosis of MS to the year of the first administration of ocrelizumab, a form of the disease (RRMS and PPMS), and the number of relapses in RRMS per year, taking into account the year before the first dose of ocrelizumab.

The clinical parameters included a collection of laboratory parameters, electrophoresis, and immunophenotyping of peripheral blood lymphocytes. The flowchart diagram (Figure 1) presents the recruitment steps and data collection procedures in a time-dependent axis.

The laboratory parameters included data on leukocytes, erythrocytes, hemoglobin, thrombocytes, neutrophils, lymphocytes, monocytes, eosinophils, basophils, aspartate aminotransferase, alanine aminotransferase, gamma-glutamyl transferase, alkaline phosphatase, and lactate dehydrogenase.

The electrophoresis of serum proteins assessed the investigation of albumin, alpha-1-globulins, alpha-2-globulins, beta-globulins, gamma-globulins, and albumin/globulin.

Immunophenotyping of peripheral blood lymphocytes included investigation of T cells, T-helper cells, T-suppressor/cytotoxic cells, ratio CR4+/CD8+, B, and NK cells.

The laboratory results in the present study were analyzed up to a maximum of 2 days before the first infusion of the first dose and one month after the second infusion of the first dose of ocrelizumab (Figure 1). The results of electrophoresis and immunophenotyping were analyzed before the first dose of ocrelizumab and before the second dose of ocrelizumab (Figure 1). All laboratory parameters were performed in the Central Laboratory of the University Hospital of Split (Department of Medical Laboratory Diagnosis, University Hospital Center Split), with the hematology analyzer (Advia 120, Siemens Bayer, Erlangen, Germany), using standardized and well-established methods, and not individually by research stuff (co-authors of the paper). Electrophoresis of serum proteins was performed by capillary electrophoresis (Capillarys 2, Sebia Hydrasys, Sebia, Camberley, UK). Immunophenotyping was performed on a FACS-Calibur flow cytometer (Becton Dickinson, Franklin Lakes, NJ, USA) using a standardized immunofluorescence method using fluorochrome-labeled monoclonal antibodies.

### 2.3. Data Analysis

The Statistical Package application for Social Science for the PC interface Windows 10 (IBM SPSS software, version 28 (IBM, Armonk, NY, USA)) was used for data analysis. Nominal variables are expressed as numbers or percentages. Numerical variables are presented as the arithmetic mean ± standard deviation and as median and interquartile range (IQR). Furthermore, the latter were compared with the Mann-Whitney U test using comparative statistics methods. The limit of statistical significance of the results was set at *p* < 0.05.

## 3. Results

The study included 15 women (68.2%) and seven men (31.8%) treated with ocrelizumab (Table 1). Out of the 22 MS patients who received the first dose of ocrelizumab, 12 had RRMS (54.55%), and 10 had PPMS (45.45%). Of the 12 patients who had RRMS, four patients relapsed once, five patients twice, and three patients three or more times within 12 months until the first administration of ocrelizumab. The median age of the subjects (*n* = 22) at the time of the first administration of ocrelizumab was 46.50 years, ranging from 38 to 53 years (Table 1). The median EDSS score was 4,00. The average duration of the disease from the diagnosis of RRMS to the first administration of ocrelizumab was 9 ± 3.46 years and 4.25 ± 4.10 years from the diagnosis of PPMS to the first administration of ocrelizumab (Table 1). Most pwMS belonged to the age group of 40 to 60 years (63.64%) (Figure 2).

There were no statistically significant differences in the laboratory parameters (Table 2), as well as no statistically significant difference in the amount of albumin and globulin, nor in the albumin–globulin ratio (Table 3) comparing the findings up to two days before and one month after the administration of the first ocrelizumab dose (*p* > 0.05).

The results of immunophenotyping of T-lymphocytes, B-lymphocytes, and NK cells were analyzed in 20 patients before the first dose and 20 patients before the second dose of ocrelizumab (Table 4, Figure 3). No statistically significant difference was found in the decrease in the value of T, T- helper, and T-suppressor/cytotoxic lymphocytes (*p* > 0.05) (Figure 3). A decrease in the values of CD19+ B-lymphocytes was found before the administration of the second dose of ocrelizumab compared to the values of CD19+ B-lymphocytes before the administration of the first dose (*p* < 0.001) (Figure 3). A statistically significant difference in the number of NK cells (CD3-CD16+56+) was not observed before the first and second administration of ocrelizumab (*p* = 0.151) (Figure 3).

## 4. Discussion

The present retrospective study compared a collection of laboratory parameters, electrophoresis, and immunophenotyping of peripheral blood lymphocytes in pwMS before intake of the first dose of ocrelizumab and before the second administration dose of ocrelizumab, which was presented in the flowchart diagram depicting the study procedure steps (Figure 1). The overall study findings report a significant depletion in B-lymphocytes (CD19+ B-lymphocytes) before the intake of the second dose of ocrelizumab in pwMS treated with ocrelizumab in a three-year period (2019–2021) at the University Hospital of Split, Croatia.

The present study findings correspond to the previously published data on the effect of ocrelizumab in pwMS regarding the inclusion of the majority of women participants, which can be attributed to the higher incidence of women suffering from MS, and consequently to the higher incidence of women receiving ocrelizumab [36,54]. The average duration of the disease from the diagnosis of MS to the first administration of ocrelizumab in the NEDA study was 4.1 ± 5.1 years for RRMS [37], and in the ORATORIO study 2.9 ± 3.2 years for PPMS patients [45]. The present study shows that somehow a longer period of time was required for the administration of ocrelizumab in RRMS patients compared to PPMS patients; the average duration for RRMS patients was 9 ± 3.46 years, and for PPMS patients, 4.25 ± 4.10 years. It is possible that the reason for the later use of ocrelizumab in the present study is that the patients received drugs from the first or second line of therapy for a longer period of time and maintained remission and minimal progression of symptoms and were switched to ocrelizumab due to MS reactivation. Considering the year of MS diagnosis and the time period from diagnosis to the administration of ocrelizumab, the present study and previously reported studies [3,41] are in accordance with the latest epidemiological data from 2022, according to which MS is diagnosed between the ages of 20 and 50 years of age, and the administration of the first dose of ocrelizumab starts a few years later after MS diagnosis [4].

Leukocytes and neutrophil granulocytes showed a minimal decrease without statistically significant results, of which three patients developed a mild form of neutropenia (up to 1.0 × 10^9^ cells/L) similar to the findings of Marodann et al. [47]. Demir et al. [55] presented the results of liver enzymes in 51 pwMS before the first infusion of the first dose of ocrelizumab and one month after the second infusion of the drug showing minimal differences in aspartate aminotransferase and gamma-glutamyl transferase enzymes without statistical significance, which suggested that ocrelizumab does not have a hepatotoxic effect after the first dose. Similar results were obtained in the present study regarding liver enzymes without statistical significance and minimal difference in aspartate aminotransferase, alanine aminotransferase, and gamma-glutamyl transferase enzymes.

Regarding the previous finding on immunophenotyping of lymphocytes in pwMS administering ocrelizumab drug, Demir et al. [55] and Capasso et al. [40] compared the total number of lymphocytes in the peripheral blood before the first infusion and one month after the second infusion of the first dose of ocrelizumab, showing a statistically significant decrease in the total number of lymphocytes, which was associated with the effect of ocrelizumab for at least another month after its administration. Compared to the findings of Capasso et al. [40] and Demir et al. [55], the present study yielded a non-significant decrease in lymphocytes in peripheral blood, which might be explained by a smaller sample of pwMS in the present study (*n* = 22) compared to the number of pwMS in Demir et al.’s (*n* = 51) and Capasso et al. (*n* = 37).

Further, comparing the immunophenotyping of T-lymphocytes before the first and the second dose of ocrelizumab, the present study showed a decrease in CD3+ T- lymphocytes, helper CD4+ T-lymphocytes, and suppressor CD8+ T-lymphocytes prior to the second administration dose of ocrelizumab, however without reaching statistical significance. The present study data coincided with the study by Cellerino et al. [3], detecting a decrease in CD3+ T- lymphocytes, helper CD4+ T-lymphocytes, and suppressor CD8+ T-lymphocytes, without statistical significance comparing the infusion of ocrelizumab before and after the first administration of ocrelizumab. The results suggest that ocrelizumab has a limited effect on T-lymphocyte depletion [3]. However, Gingele et al. [41] reported a significant decrease in CD20+ T- lymphocytes after the administration of the first dose of ocrelizumab. As CD20+ T-lymphocytes make up 18.4% of the total number of CD20+ cells, these results suggest that ocrelizumab is not exclusively specific for B- lymphocytes and that CD20+ T-lymphocytes potentially play a role in the immunopathogenesis and reduced activity of MS [41,42]. We were not able to perform immunophenotyping of CD20+ T-lymphocytes due to technical reasons, but the results of the previous studies could explain the decrease in the number of T-lymphocytes observed in the present study. It is possible that pwMS have a slightly higher number of CD20+ T-lymphocytes compared to the healthy population, but further studies are needed to investigate this hypothesis.

The analysis of CD19+ B-lymphocytes is usually controlled because the presence of the drug can interfere with the analysis of CD20+ B- lymphocytes [39], and the CD19+ marker is present in all cells of the B lineage in humans [56]. By performing immunophenotyping of lymphocytes before the first and before the second dose of ocrelizumab, we observed the expected drastic decrease in CD19+ B cells. Immunophenotyping of lymphocytes in pwMS who received ocrelizumab was measured between weeks 8th and 24th after the first dose of the drug. Margoni et al. [39] state that the highest depletion of B-lymphocytes after the administration of ocrelizumab occurs after the 8th week from the moment of receiving the dose of the drug, and this lowest value of B- lymphocytes can last for months, which is in accordance with our results. Abbadesse et al. [33] included a large sample of 218 pwMS and reported a statistically significant decrease in CD19+ B- lymphocytes. Fernández-Velasco et al. [57] included 53 pwMS with PPMS in a multicentric study and reported a significant decrease in the CD19+ B cells at six months of ocrelizumab treatment. The results of our present study correspond to the previously published data [33,39,57] and confirm the primary action of ocrelizumab on the B cell lineage.

Regarding the NK cells, Fernández-Velasco et al. [57] observed a decrease in CD56+ NK cells comparing immunophenotyping before and after the first dose of ocrelizumab, but these results did not reach statistical significance. Abbadessa et al. [51] showed a minimal decrease in NK cells without statistical significance. The present study’s findings are comparable with the aforementioned studies’ findings concerning NK cells [51,57]. Perini et al. [31] analyzed NK cells before and after interferon beta administration and found that the drug acts on the depletion of CD3-CD16+ and CD3-CD57+ NK cells, while Al-Falahi et al. [25], in animal models with experimental autoimmune encephalomyelitis, showed that glatiramer-acetate has a much stronger killing efficiency of one’s own NK cells against dendritic cells. Potentially, ocrelizumab acts cumulatively with the remnants of previously received drugs and reactivates their effect on NK cell depletion by some unknown mechanism. CD3- CD16+CD56+ is a biomarker for cytotoxic NK cells that act to destroy cells infected with viruses, affected by cancer, or responsible for tissue rejection after transplantation.

The lower level of CD3-CD16+CD56+ NK cells could make pwMS extremely sensitive to the development of herpes infections [58], and in our study one patient developed a herpes virus infection after ocrelizumab administration. The possibility of the appearance of malignant diseases after the use of ocrelizumab has also been described, which is higher compared to the placebo and compared to the group that received interferon beta [46] and could be explained, apart from the reduced immune response due to the depletion of the B cell line, and the reduced immune response due to depletion of CD3-CD16+CD56+ NK cells. Given that ocrelizumab is relatively a new drug (approved by Food and Drug Administration on 28 March 2017, in the USA) [35,59,60], with a partially insufficiently investigated immunopathogenesis mechanism, it is yet not known how the drug acts on CD3-CD16+CD56+ NK cells. The analysis of side effects after the administration of ocrelizumab is beyond the scope of this study, and further studies are needed to fully confirm the immunopathogenesis of action of the drug. The effect of the first dose in the present study was important in order to confirm the effectiveness of the drug, as well as to satisfy the conditions for the continuation of its use while at the same time achieving safety for our patient because a better understanding of the dynamics of the lymphocyte count allows for an optimal balance of the effectiveness and safety of the drug. The future study is suggested to follow-up the pwMS after the second ocrelizumab dose.

## 5. Conclusions

A significant depletion in B-lymphocytes (CD19+ B-lymphocytes) was evident before the intake of the second dose of ocrelizumab in pwMS treated with ocrelizumab, confirming the previous findings [33,39,57] on the primary action of ocrelizumab on the B cell lineage.

## Figures and Tables

**Figure 1 biomedicines-11-00353-f001:**
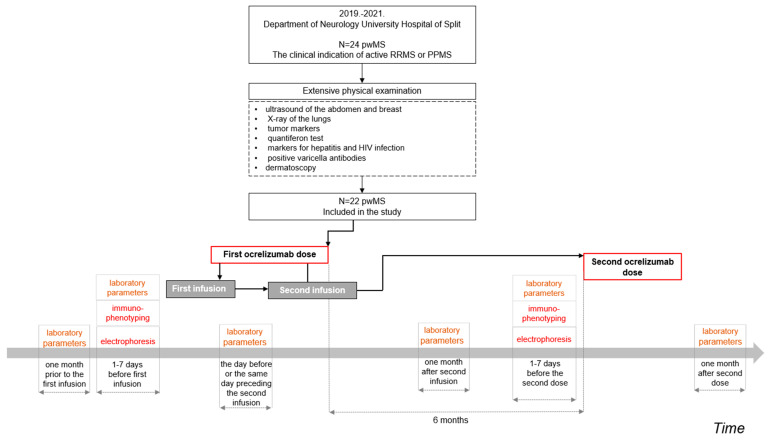
The flowchart diagram depicts the recruitment of pwMS and the study data collection procedure. The clinical parameters analyzed in the present study included: (a) laboratory parameters prior to the first infusion of the first ocrelizumab dose and one month after the second infusion of the first ocrelizumab dose, (b) electrophoresis prior to the first infusion of the first ocrelizumab dose and prior to the second ocrelizumab dose, and (c) immunophenotyping prior to the first infusion of the first ocrelizumab dose and prior to the second ocrelizumab dose. Note: laboratory parameters were clinically collected more frequently.

**Figure 2 biomedicines-11-00353-f002:**
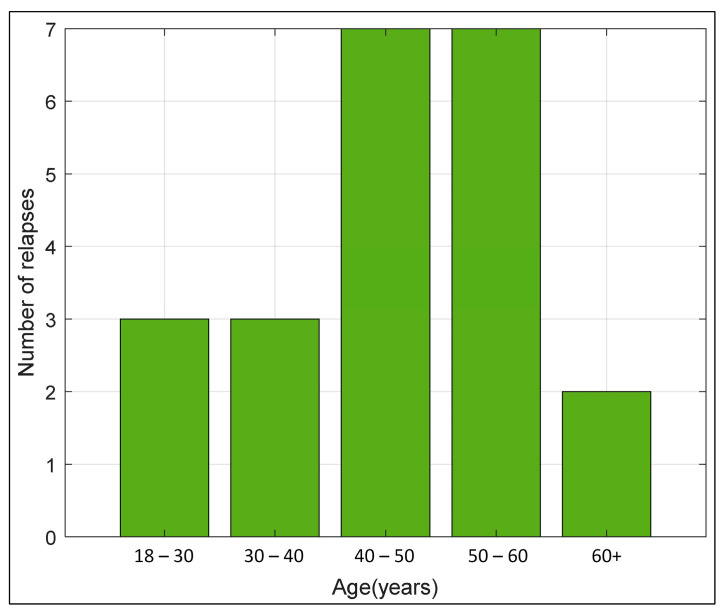
Distribution of patients treated with ocrelizumab by age group.

**Figure 3 biomedicines-11-00353-f003:**
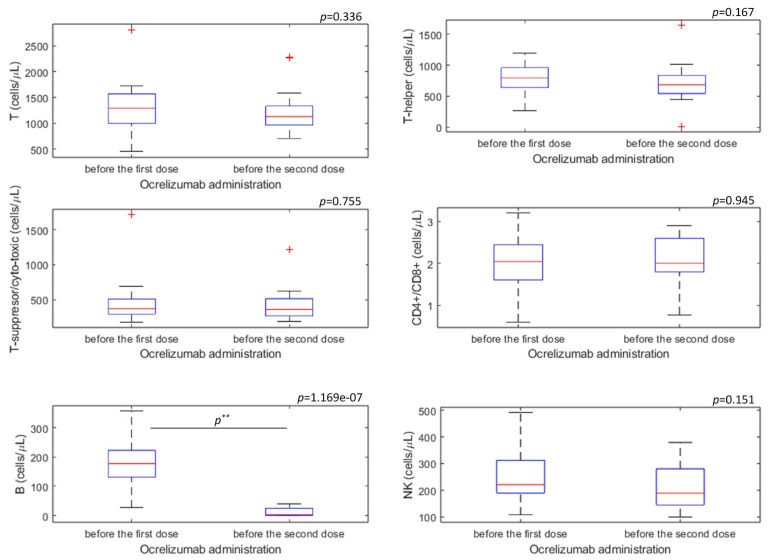
Graphical representation results of immunophenotyping of T-lymphocytes, B-lymphocytes, and NK cells before the first and the second dose of ocrelizumab administration in pwMS after applying Mann-Whitney U-test. Note: On each box, the central red mark indicates the median, and the bottom and top edges of the box indicate the 25th and 75th percentiles, respectively. The whiskers represent data between the minimum and maximum values not considered outliers, and the outliers are plotted individually using the red ‘+’ marker symbol. *p* ** < 0.001.

**Table 1 biomedicines-11-00353-t001:** Demographic and disease-related data.

Age of patients at the start of ocrelizumab administration (years)	*n* = 22 46.50 IQR (38. 53) min-max (21–67)
EDSS score before the first administration of ocrelizumab EDSS score before the second administration of ocrelizumab	*n* = 22 4.00 (2.63–5.38) 4.00(2.65–5.38)
Duration of RRMS disease at the first administration of ocrelizumab (years)	*n* = 9 9 ± 3.46
Duration of PPMS at the first administration of ocrelizumab (years)	*n* = 8 4.25 ± 4.1

Duration of the disease from the diagnosis of multiple sclerosis to the first administration of ocrelizumab is expressed as mean, and the EDSS score is presented as the median and interquartile range. RRMS = relapsing-remitting multiple sclerosis; PPSM = primary progressive multiple sclerosis. IQR—InterQuartile Range.

**Table 2 biomedicines-11-00353-t002:** Laboratory parameters before administration of the first infusion of the first dose and one month after the administration of the second infusion of the first dose of ocrelizumab.

Variable	Before Administration of the First Ocrelizumab Infusion	One Month after the Second Ocrelizumab Infusion	*p* *
Leukocytes (×10^9^/L)	(*n* = 21) 6.00 (4.90–7.00)	(*n* = 20) 5.75 (5.25–7.20)	0.938
Erythrocytes (×10^12^/L)	(*n* = 21) 4.80 (4.31–4.93)	(*n* = 20) 4.78 (4.42–4.91)	0.927
Hemoglobin (g/L)	(*n* = 21) 139.00 (128.00–149.00)	(*n* = 20) 138.00 (130.75–148.25)	0.927
Thrombocytes (×10^9^/L)	(*n* = 21) 233.00 (221.00–284.00)	(*n* = 20) 251.50 (215.75–279.50)	0.764
Neutrophils (×10^9^/L)	(*n* = 21) 3.70 (2.68–4.47)	(*n* = 20) 3.40 (3.20–4.98)	0.676
Lymphocytes (×10^9^/L)	(*n* = 21) 1.59 (1.24–2.10)	(*n* = 20) 1.36 (1.18–1.71)	0.100
Monocytes (×10^9^/L)	(*n* = 21) 0.41 (0.33–0.53)	(*n* = 20) 0.46 (0.40–0.57)	0.347
Eosinophils (×10^9^/L)	(*n* = 21) 0.14 (0.10–0.22)	(*n* = 20) 0.13 (0.09–0.20)	0.657
Basophils (×10^9^/L)	(*n* = 21) 0.04 (0.03–0.05)	(*n* = 20) 0.04 (0.03–0.05)	0.594
AST (U/L)	(*n* = 19) 19.00 (18.00–21.50)	(*n* = 20) 20.00 (16.75–23.00)	0.592
ALT (U/L)	(*n* = 19) 20.00 (17.50–28.00)	(*n* = 20) 19.50 (16.00–24.25)	0.663
GGT^c^ (U/L)	(*n* = 20) 15.00 (12.00–22.00)	(*n* = 20) 15.00 (12.75–21.00)	0.881
ALP (U/L)	(*n* = 7) 61.00 (49.00–74.00)	(*n* = 8) 59.50 (51.00–73.50)	0.862
LDH (U/L)	(*n* = 13) 158.00 (142.00–168.00)	(*n* = 11) 178.00 (149.50–181.50)	0.401

Data are presented as median and interquartile range. AST = aspartate aminotransferase; ALT = alanine aminotransferase; GGT = gamma-glutamyl transferase; ALP = alkaline phosphatase; LDH = lactate dehydrogenase. * Mann-Whitney U test.

**Table 3 biomedicines-11-00353-t003:** Serum protein electrophoresis values before the first and before the second dose of ocrelizumab.

Electrophoresis	Before Administration of the First Dose of Ocrelizumab (*n* = 16)	Before Administration of the Second Dose of Ocrelizumab (*n* = 15)	*p* *
Albumin (g/L)	43.70 (41.14–44.30)	42.14 (41.20–43.15)	0.265
Alpha-1-globulins (g/L)	2.68 (2.50–3.00)	2.80 (2.69–2.98)	0.635
Alpha-2-globulins (g/L)	6.51 (6.19–7.34)	6.60 (6.30.6.94)	0.797
Beta-globulins (g/L)	7.20 (6.90–8.13)	7.42 (7.02–7.89)	0.751
Gamma -globulins (g/L)	9.78 (8.64–10.88)	9.59 (8.55–10.48)	0.89
Albumin/globulins	1.63 (1.53–1.71)	1.62 (1.50–1.71)	0.736

Data are presented as median and interquartile range. * Mann-Whitney U test.

**Table 4 biomedicines-11-00353-t004:** Results of immunophenotyping of T-lymphocytes, B-lymphocytes, and NK cells.

Immunophenotyping of Lymphocytes	Cell Marker	Before Administration of the First Dose of Ocrelizumab (*n* = 17)	Before Administration of the Second Dose of Ocrelizumab (*n* = 19)	*p* *
T (cell/µL)	CD3+	1287.00 (1004.00–1560.75)	1129.50 (972.00–1329.25)	0.336
T-helper (cell/µL)	CD3+CD4+	790.00 (656.50–937.50)	685.00 (560.00–829.50)	0.167
T-suppressor/cytotoxic(cell/µL)	CD3+CD8+	367.50 (298.25–509.00)	357.00 (264.50–495.50)	0.755
Ratio CD4+/CD8+	CD4+/CD8+	2.05 (1.60–2.38)	2.00 (1.80–2.60)	0.946
B (cell/µL)	CD19+	177.50 (134.50–220.25)	3.50 (1.00–23.25)	<0.001 *
NK (cell/µL)	CD3− CD16+ 56+	221.00 (189.75–299.00)	189.00 (146.25–266.75)	0.151

Data are presented as median and interquartile range; * Mann-Whitney U test; natural killer (NK) cells.

## Data Availability

The data can be provided upon request to the authors.

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
