# Peer review of "Confirmation of CD19+ B-Lymphocyte Depletion Prior to Intake of the Second Dose of Ocrelizumab in Multiple Sclerosis Patients"

_biomedicines, 2023, doi:10.3390/biomedicines11020353_

Round 1
Reviewer 1 Report
The manuscript titled “Confirmation of CD19+ B-lymphocyte depletion prior to intake of the second dose of ocrelizumab in multiple sclerosis patients” describes the effects of a single dose of ocrelizumab in MS patients and provides confirmation of what is seen in other MS studies. A single dose of ocrelizumab significantly depletes CD19+ B cells without having much effect on other cell populations (T cells, NK cells etc.) Though this study does not provide novel data, it is important to confirm other clinical studies using different populations. Additionally it suggests the off-target effects of the drug on T cells may not be seen in everyone. However, CD20+ T cells were not directly assessed.
Few minor comments:
1. Clearly the title argues that the goal is to look at the effects of ocrelizumab after the first dose, but why not also assess if there are changes after the second dose? If this data is readily available, is it worth including? If not, please add reasoning to the discussion or introduction for only focusing on the effects of the first dose.
2. More information is needed in the methods. Are all the laboratory parameters run in clinical labs? Or by the authors themselves? Meaning if the data are from CBCs and CBCs with differential please indicate as such. Same for the electrophoresis data are these in house of clinical assays. If flow cytometry was used for the immunophenotyping data then that needs to be included as well as representative gating strategies.
Author Response
Attached is the response to Reviewer 1 comments.

Reviewer 2 Report
The authors investigated immunophenotyping of T-lymphocytes, B-lymphocytes, and natural killer cells in 22 patients with multiple sclerosis before administration of the first and second dose of ocrelizumab. A significant decrease in the number of CD19-positive B-lymphocytes, but not in those of T-lymphocytes or natural killer cells, was observed before administration of the second dose of ocrelizumab.
This is an interesting study providing important insights into current knowledge on the management of multiple sclerosis. Taking up the topic of multiple sclerosis is timely because new therapeutic options for this disease now appear one after another.
Although I do not have any critical comments, minor issues and suggestions to strengthen this manuscript are raised as follows:
1. Please reconfirm the use of abbreviations. For example, “NK” and “pwMS” in the abstract are not needed because “natural killer” and “patients with multiple sclerosis” appear only once in here.
2. Abbreviations in tables, such as “RRMS”, “PPMS” “AST”, “ALT”, and “GGT”, should be explained in the footnote of tables.
3. Although the therapeutic strategy targeting B-lymphocytes has recently attracted researchers, multiple sclerosis was traditionally considered a disease mediated by T cells, particularly CD4‐positive T cells reactive to myelin antigens. Additionally, increasing evidence suggests that macrophages derived from circulating monocytes and resident microglia play a pivotal role in the pathogenesis of multiple sclerosis. I would suggest incorporating these historical backgrounds to facilitate comprehension, by citing a relevant article describing these issues (e.g., Cells 2021; 10: 844).
Author Response
Attached is the response to Reviewer 2 comments.
